# A Case-Study Application of the Experimental Watershed Study Design to Advance Adaptive Management of Contemporary Watersheds

**Jason A. Hubbart** [1,*] , **Elliott Kellner** [2] **and Sean J. Zeiger** [3]

1    West Virginia University, Institute of Water Security and Science, Davis College of Agriculture,
     Natural Resources and Design, Schools of Agriculture and Food, and Natural Resources, 3109 Agricultural
     Sciences Building, Morgantown, WV 26506, USA

2    West Virginia University, Institute of Water Security and Science, Davis College of Agriculture Natural
     Resources and Design, Division of Plant and Soil Sciences, 3011 Agricultural Sciences Building,
     Morgantown, WV 26506, USA; Elliott.Kellner@mail.wvu.edu

3    School of Natural Resources, University of Missouri, 203-T ABNR Building, Columbia, MO 65211, USA;
     sjzxbb@mail.missouri.edu

*    Correspondence: Jason.Hubbart@mail.wvu.edu; Tel.: +1-3042932472

**Abstract:** Land managers are often inadequately informed to make management decisions in contemporary watersheds, in which sources of impairment are simultaneously shifting due to the combined influences of land use change, rapid ongoing human population growth, and changing environmental conditions.    There is, thus, a great need for effective collaborative adaptive management (CAM; or derivatives) efforts utilizing an accepted methodological approach that provides data needed to properly identify and address past, present, and future sources of impairment. The experimental watershed study design holds great promise for meeting such needs and facilitating an effective collaborative and adaptive management process. To advance understanding of natural and anthropogenic influences on sources of impairment, and to demonstrate the approach in a contemporary watershed, a nested-scale experimental watershed study design was implemented in a representative, contemporary, mixed-use watershed located in Midwestern USA. Results identify challenges associated with CAM, and how the experimental watershed approach can help to objectively elucidate causal factors, target critical source areas, and provide the science-based information needed to make informed management decisions. Results show urban/suburban development and agriculture are primary drivers of alterations to watershed hydrology, streamflow regimes, transport of multiple water quality constituents, and stream physical habitat.    However, several natural processes and watershed characteristics, such as surficial geology and stream system evolution, are likely compounding observed water quality impairment and aquatic habitat degradation.    Given the varied and complicated set of factors contributing to such issues in the study watershed and other contemporary watersheds, watershed restoration is likely subject to physical limitations and should be conceptualized in the context of achievable goals/objectives.    Overall, results demonstrate the immense, globally transferrable value of the experimental watershed approach and coupled CAM process to address contemporary water resource management challenges.

**Keywords:** urban watershed management; municipal watershed; water quality impairment; collaborative adaptive management; water resources; urban watersheds

## 1. Challenges in Contemporary Watershed Management

### 1.1. Collaborative Adaptive Management

Contemporary watershed management problems are complex, consisting of multiple, conflicting, and non-linear and/or stochastic variables. Research has shown that the management of watersheds is most effective using an adaptive and integrated approach based on iterative applications of best (or better) practices guided by ecosystem process responses. Adaptive management comprises critical steps that include (but are not limited to) problem assessment, remediation design, implementation, monitoring, evaluation, and management plan adjustment [1–5]. Based on initial problem assessments, a project is often designed and implemented, and then, with regular monitoring and (re)evaluation, adjustments may be applied, and projects revised. This iterative process helps update management plans over time while incorporating additional precautions and experiences garnered from new information. In addition to complexity and uncertainty, natural resource management is interconnected through equally complex and intermingled land use needs (practices) of humans. Land and water management, therefore, benefit from a collaborative approach that includes multiple stakeholders. Collaborative Adaptive Management (CAM) facilitates the introduction of local stakeholders as a major component of sustainable decision-making. It is recognized that there may be other approximate derivatives to CAM, but for simplicity, here we reference CAM. The primary goal of CAM is to integrate knowledge and science with experience and the perspectives of scientists, stakeholders, and managers for more effective management decision-making [6–9].

Collaborative Adaptive Management has been applied broadly in landscape-level planning and management globally [10]. There have been various applications of the collaborative adaptive process in regions of the world including (but not limited to) Southeast Asia [11], Brazil [12,13], and Europe [14–20]. Most CAM applications have been in the United States and Australia [21–23]. One such application in the United States is the Chesapeake Bay Program (CBP), a globally recognized model for the collaborative management and restoration of large aquatic ecosystems [24]. The Chesapeake Bay Total Maximum Daily Load (TMDL) [25], administered by the U.S. Environmental Protection Agency (EPA), is one of the largest and longest-running pollution control programs in history. The Chesapeake Bay Watershed Agreement (2014) was drafted based on contributions from numerous federal and state agencies, citizens, stakeholders, academic institutions, local governments, and non-profit organizations [26]. The regulatory program is currently implemented in six states (Delaware, Maryland, New York, Pennsylvania, Virginia, West Virginia), and the District of Columbia, and mandates reductions of three primary constituents of concern (i.e., nitrogen, phosphorus, and suspended sediment) to improve various indicators of Bay-water quality and aquatic habitat (e.g., dissolved oxygen, turbidity, submerged aquatic vegetation). The Chesapeake Bay Program provides a model for CAM activities, including stream restoration, upland pollutant source reduction, infrastructure improvements such as urban green streets, and retrofitting existing stormwater facilities to improve water quality [27]. Another leading example of CAM in the United States is the Mississippi River Basin, Gulf Hypoxia Program coordinated by the Mississippi River/Gulf of Mexico Watershed Nutrient Task Force (Hypoxia Task Force or HTF). The HTF is a collaborative management effort by state and federal agencies established, at least in part, to reduce the size and persistence of "The Dead Zone", a large area of hypoxia (i.e., oxygen concentration less than 2 mg L$^{-1}$) in the Gulf of Mexico [28–32]. The CAM model has been similarly utilized to improve the management of various natural resources, including (but not limited to) wild fisheries management [33], surface and groundwater resource allocation [34], and urban water management [35] in Australia. Collectively, action plans associated with CAM programs highlight the importance of accounting for future monitoring information, changing environmental conditions, and lessons learned globally [21,31,32], thereby emphasizing the need for high-quality environmental data to inform effective management of natural resources.

### 1.2. Environmental Monitoring to Improve Management

Over 100 years ago, watershed managers recognized the need to better understand land use and water quality and quantity relationships, in order to improve management practices and stewardship and to sustain natural resource commodities. There was an urgency to understand how the water balance of a given watershed is controlled by climate, soil, and vegetation interactions, and how alterations of such factors may affect the water regime (i.e., timing and quantity of water), water quality, and various related natural resources [36,37]. Among the first studies in the United States to address nationwide watershed issues was the Wagon Wheel Gap Experimental Watershed Study, which started in 1909 to protect navigable streams at the watershed scale [38,39]. Other early studies focused on the effects of road building and forest harvest practices on water quantity and quality (e.g., flow and flow velocity, erosion, sediment, nutrients). Later studies included agricultural impacts. However, despite advancements generated through early studies, watershed mismanagement continues to be identified as primarily responsible for anthropogenic disturbances of waterways [40–43], and land managers remain poorly equipped to address contemporary mixed-land-use watershed issues that are set in a continuum of forested, agriculture, *and* urban land use types and are associated with aggressive human population growth. For context, a recent report published by the United Nations [44] showed approximately 30% of the global human population (751 million people) lived in metropolitan areas during 1950. By 2018 that percentage had grown to 55% (4.2 billion urban inhabitants) [44]. By 2050, it is projected that almost 70% of the global human population, approximately 6.7 billion people, will live in metropolitan areas [44]. Moreover, there are concomitant, growing, human health and quality-of-life issues related to water resources that are global in scale. There are increasing demands for management solutions and guarantees of sustainable water resources and water quality for future generations, which will depend on research, education, outreach, collaboration, adaptive management, and understanding the cultural anthropology of water [45]. Considering the scope of these complexities, is there any question that we must reconsider all that we think we know, and reimagine watershed management, given the rapid succession of intermingled impacts in recent decades alone?

### 1.3. Contemporary Application of the Experimental Watershed Approach

The nested-scale and/or paired experimental watershed study designs (and other derivations) have been shown to be effective approaches for quantitatively characterizing hydrologic and water quality perturbations in mixed-land-use watersheds [46–57]. Nested watershed study designs utilize a series of sub-catchments inside a larger watershed to monitor land use impacts on environmental variables of interest. A paired watershed study design includes data collection from at least two watersheds (control and treatment) with similar physiographical characteristics. Sub-catchments are delineated to isolate land use types and hydrologic characteristics. While often applied at the watershed scale, the design concept can be applied at any scale, from the reach level (scale-nested) to the basin level. Ultimately, the design enables researchers to partition and quantify the influencing processes observed at the sub-catchment scale [58], and thereby determine the influence and cumulative effect of dominant land use types on the response variable of interest. By applying a nested-scale experimental watershed approach, factors (e.g., land use, hydroclimate) contributing to a given variable of interest may be more effectively (objectively) disentangled, producing quantitative information regarding hydrologic and water quality regimes related to specific land-uses. For example, Tetzlaff et al. [57] discussed how the experimental watershed approach has yielded several benefits, including (but not limited to) (1) science-based information to answer site-specific management questions, (2) quantitative information needed for ongoing model development, and (3) ground-truthing of large-scale remote sensing data. Experimental watershed studies can elucidate unknown problems in a watershed of interest, where unique combinations of natural and anthropogenic (legacy and ongoing) conditions would otherwise confound planning efforts [59]. Additionally, experimental watershed studies can enable the discovery of previously unknown phenomena and/or processes that contribute to globally important natural resources security issues. For example, results from Hubbard Brook Experimental Forest involving

the acid rain phenomenon were especially transformative, with important global implications for natural resources management [48]. A study by Felson and Pickett [60] showed that scientists and urban designer partnerships could result in a deeper understanding of urban land-use influence on ecological response across spatial scales, and rural-urban land use gradients [57]. Such information is important considering the combined influence of hydroclimate extremes and land use change that is expected to continue to degrade water resources and ecological health in future decades [41].

Despite the potential for experimental watershed studies to yield valuable information for land and water resource managers [59], the approach is rarely applied in contemporary, mixed-land-use watersheds due to seemingly daunting challenges. Felson and Pickett [60] noted challenges associated with collaborative urban planning, and science-based experimental design efforts include, (1) the need for enhanced communication between planners, scientists, and stakeholders, (2) a lack of control over experimental installation during initial urban planning and development, and (3) associated costs and financial limitations typically faced by local municipalities. However, given the increasing rate and intensity of global water resource degradation, effective methods must be utilized to overcome obstacles to implementation, regardless of the level of complexity. Therefore, while the challenges noted by previous authors are certainly affirmed, investment costs in the shorter term might be outweighed by long-term irretrievable effects of less informed management. Contemporary application of the experimental watershed approach is an effective method that can provide the detailed information necessary to improve management, conservation, and sustainability of water resources while driving down long-term costs. To date, one of the few examples of a mixed-land-use, contemporary experimental watershed study is Hinkson Creek Watershed [53]. The purpose of this article is to provide an example of the successful integration of the experimental watershed study design and collaborative adaptive management to advance policy and management practices in contemporary watersheds.

## 2. Case Study: Hinkson Creek Watershed

### 2.1. Case Study Setting

To provide important context for the reader, in particular, pertaining to transferability to other mixed (multi) use watersheds globally, we provide information about the case-study watershed used for the current work as follows. Hinkson Creek Watershed (HCW) is located within the Lower Missouri-Moreau River Basin (LMMRB) in central Missouri, USA (Figure 1) [53]. The main channel, Hinkson Creek, is a 3rd order stream that flows through a basin of approximately 231 km$^2$ [53]. At the time of this work, urban areas of HCW were primarily residential with progressive commercial expansion from the City of Columbia (population approximately 122,000) [61]. Land use in the watershed was approximately 32% forest, 37% pasture or cropland, and 29% urban (Table 1). The regional climate is dominated by continental polar air masses and maritime and continental tropical air masses during the winter and summer, respectively. The mean annual total precipitation is approximately 1096 mm, and the mean annual air temperature is approximately 13.5 °C. A wet season occurs primarily from March through June. A portion of the LMMRB was targeted as critical for controlling erosion and nonpoint source pollution in 1998 [53]. Watershed restoration efforts in the LMMRB were accelerated by mandates of the Clean Water Act (CWA) and subsequent lawsuits. Hinkson Creek Watershed is representative of the LMMRB, and many developing watersheds globally, with respect to hydrologic processes, water quality, climate, and land use. Similar to many watersheds, the impaired use for Hinkson Creek was identified as "protection of warm water aquatic life" from unknown pollutants [62].

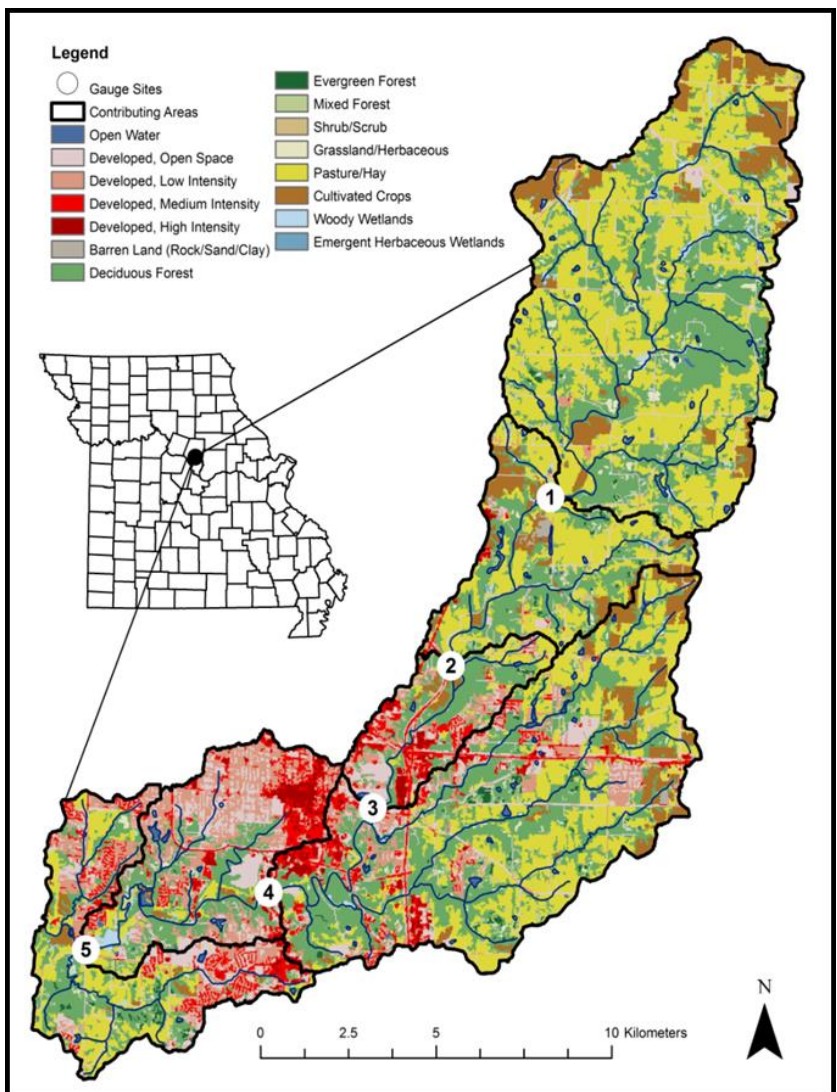

**Figure 1.** Locations of gauge sites (where #4 includes the USGS gauging station) and corresponding drainage area to each gauge (bold line) in the Hinkson Creek Watershed (HCW), in Central Missouri, USA. A model urban nested-scale experimental watershed study.

**Table 1.** Cumulative land use and land cover (LULC), drainage area, and stream length corresponding to each gauging site located in Hinkson Creek Watershed (HCW), Missouri, USA. Percent cumulative LULC is shown parenthetically.

| Variable [km² (%)] | Site #1 | Site #2 | Site #3 | Site #4 | Site #5 | HCW |
|---|---|---|---|---|---|---|
| Agricultural | 45.0 (57.0) | 56.4 (54.9) | 57.6 (49.5) | 78.5 (43.1) | 79.7 (38.4) | 85.4 (36.7) |
| Forested | 28.4 (35.9) | 37.5 (36.4) | 41.1 (35.4) | 62.8 (34.5) | 68.6 (33.1) | 74.9 (32.2) |
| Urban | 3.7 (4.7) | 6.6 (6.4) | 15 (13.0) | 37.1 (20.4) | 54.9 (26.5) | 67.6 (29.0) |
| Wetland | 1.9 (2.4) | 2.4 (2.3) | 2.5 (2.1) | 3.6 (2.0) | 4.4 (2.0) | 4.9 (2.1) |
| Total area | 79.0 | 102.9 | 116.2 | 182.0 | 207.5 | 232.8 |
| Stream length [§] | 22.8 | 29.8 | 35.4 | 43.6 | 53.0 | 56.1 |

[§] Stream length is shown in km.

Hinkson Creek's listing on the Clean Water Act (CWA) 303(d) list as impaired by unknown pollutants in 1998 [59] came about due to many issues identified by state and federal agencies and local residents, including (but not limited to), (1) larger and more frequent floods; (2) lower base flows; (3) increased soil erosion in construction and development areas with subsequent transport

of the soil to streams (i.e., altered suspended sediment regimes); (4) water contamination from urban stormwater flows; (5) degradation of habitat for aquatic organisms due to the concerns listed above; and (6) degradation of aquatic habitat due to the physical alteration of stream channels and streamside (riparian) corridors [55,63–69]. In 2008, the watershed was instrumented with a nested-scale experimental watershed study design [53] to generate data that address the uncertainties of the 303(d) listing, while providing a scientific basis for developing a TMDL target. The experimental watershed program was designed to investigate the problems suspected to have led to the 1998 listing and improve understanding of contemporary land-use effects on hydrologic processes (stream response, water yield), water quality, and biological community health. Each nested monitoring site in Hinkson Creek was designed to monitor water stage and a complete suite of climate variables. Multiple additional water quality variables (e.g., suspended sediment, nitrogen, phosphorus, chloride, pH, and other constituents) were monitored at the nested sites shortly after implementation of the study. A United States Geological Survey gauging station (USGS-06910230) had collected stage data intermittently since 1966 and provided flow data for site 4 (Figure 1). Articles from the Hinkson Creek Experimental Watershed (HCEW) program were being published as early as 2010. To date, there have been over 50 publications in peer-reviewed journals and 21 graduate student theses and dissertations.

### 2.2. Collaborative Adaptive Management

In 2011, a Collaborative Adaptive Management (CAM) program was developed to provide direction and support for the 303(d)-delisting process (www.helpthehinkson.org) [64]. The CAM process was designed to be fundamentally science-based as doing so acknowledges uncertainties and/or unknowns about complex systems, engages scientists, decision-makers, and stakeholders, and applies continuous process improvements to reduce those uncertainties and maximize the opportunity for success [70,71]. In this manner, a science-driven CAM process can support efforts aimed at improving water quality and aquatic habitat in contemporary watersheds, because scientific results and the understanding they foster can guide informed decision-making. This approach is important because, in complex contemporary watershed systems, applying a mitigation strategy may improve one or more characteristics of the stream, but not achieve the ultimate goal. Typically, when a stream or other water body is listed as impaired, a Total Maximum Daily Load (TMDL) analysis is conducted to define the maximum pollutant load compatible with full compliance of the stream with designated uses [72]. However, this approach can be confounded when no specific pollutant has been identified. From the outset of the regulatory process, impairment of Hinkson Creek was assumed predominantly a result of urban development. Given the listing of the creek for "unknown pollutants", a volume-based flow reduction strategy was initially adopted, which was focused on urban stormwater runoff reduction as a means to reduce unknown pollutant concentrations and loadings [53]. Specifically, a target of 50% volume reduction was set for HCW in the waste load allocation (proportion of stormwater attributed to point sources) developed by the Missouri Department of Natural Resources (MDNR) [73]. The wasteload allocation was required to be met by urban and developed areas, while the load allocation (proportion of stormwater attributed to nonpoint sources) was assigned to rural areas [53]. Such volume-based approaches are encouraged by USEPA and the National Research Council [74]; therefore, the application of stormwater reduction as a surrogate for pollutants is not uncommon [53]. These details, and those that follow, chronicle the cumulative results of research conducted within the context of the HCW study and demonstrate the immeasurable value of the experimental watershed approach and the CAM program to water resource management.

### 2.3. Experimental Watershed Design Outcomes

The experimental watershed study design applied in HCW facilitated the identification and quantification of factors contributing to impairment of the stream and provided the information needed to target mechanistic drivers, both natural and anthropogenic, of hydrologic alteration. Detail is provided here to give the reader a sense of the scope of possible findings that can be obtained via

the methodological approach. The analysis showed that annual streamflow metrics (i.e., peak flow, baseflow) had not significantly increased or decreased in Hinkson Creek from 1967 to 2010 [63]. However, more recent work indicated that significant changes in runoff volume and timing in the watershed (largely due to urbanization) have occurred in the years up to 2015 [75]. Additionally, event-based (30-min interval) rainfall-streamflow response showed increased explained variance at urban sites relative to rural sites, indicating the potential for increased streamflow response to rainfall events at urban sites [61]. Multiple ($n = 12$) event-based streamflow regime metrics (e.g., peak flow magnitude and timing), which were calculated from observed paired-independent storm events were correlated with urban land use [67]. A positive relationship between developed land uses (i.e., urban and suburban) and volumetric streamflow was consistently observed through various analyses [76–81], thus highlighting the importance of land use impacts on streamflow characteristics and sediment transport [56,59,77,80,82,83].

Suspended sediment levels in Hinkson Creek may be high for the region [82]. There was a disproportionately high contribution of fine sediment reported from the City of Columbia, relative to Hinkson Creek [84,85]. While the variability of spatiotemporal distributions of suspended sediment particle densities (e.g., organic material) in Hinkson Creek can confound loading estimations [86,87], work conclusively showed that average suspended sediment particle size decreased in Hinkson Creek as cumulative urban land use increased in the watershed. Moreover, a doubling of streamflow more than doubled (i.e., a non-linear relationship) fine suspended sediment concentrations in Hinkson Creek [88,89]. In addition, studies showed that nearly all (99%) of the total suspended sediment load was transported during high flows ($Q_{10}$) [76].

A study in 2011 showed that stream bank erosion contributed approximately 67% of suspended sediment loading over the 2011 water year, illustrating the potential contribution of in-stream vs. terrestrial suspended sediment in the watershed [90]. Kellner and Hubbart [81] showed that channel widening and incision in Hinkson Creek (e.g., erosion of streambed and banks) were spatially correlated to developed land uses, and associated streamflow characteristics, in the middle and lower watershed. Increased erosion of streambeds and banks due to urban runoff may help explain observed suspended sediment patterns and further emphasizes the importance of streamflow to sediment and pollutant transport dynamics. However, suspended sediment is only one of the set of factors influencing water quality. Alterations to multiple nutrient constituents, driven by land use practices, were observed in HCW [56,76]. Zeiger and Hubbart [56] showed total inorganic nitrogen and nitrate concentrations were relatively higher in the agricultural headwaters. Increased nitrate levels are quite common in the agricultural areas of the Midwest, particularly the Upper Mississippi River Basin, where nitrogen fertilizer applications can exceed 2.5 t $km^{-2}$ $yr^{-1}$. However, total ammonia yields greater than 1.25 kg $ha^{-1}$ $yr^{-1}$ and total phosphorus yields exceeding 2.0 kg $ha^{-1}$ $yr^{-1}$ in Hinkson Creek were high for the Mississippi River Basin [56]. Total phosphorus concentrations exceeded 1.13 mg $L^{-1}$ at suburban/urban sites.

Urban land uses also correlated with adverse physicochemical characteristics in Hinkson Creek, including toxic chloride concentrations and loadings [64], altered dissolved oxygen trends (both above and below established water quality standards [91,92]), and increased pH and total dissolved solids [83]. Hubbart et al. [64] showed chloride in Hinkson Creek reaches seasonally-mediated acute (860 mg $L^{-1}$) and chronic (230 mg $L^{-1}$) concentrations with lower concentrations persisting in floodplain shallow groundwater year-round. Collectively, the results of stream physicochemistry investigations suggest the potential for aquatic biota stress throughout the main stem of Hinkson Creek and identify land-use practices as a primary driver of water resource degradation. Results also showed that urbanization (Columbia, Missouri) has resulted in significantly ($p < 0.05$) altered stream water temperature regimes [54,93,94]. Daily maximum stream temperature exceeded a threshold of potential mortality of warm-water biota (i.e., 35.0 °C). Additionally, maximum stream temperature was 4.0 °C greater at an urban monitoring site, relative to a rural site for 10.5 h, indicating urban land use exacerbates the influence of summertime drought on thermal stream conditions. Sudden increases

in stream temperature (stream temperature surges) were observed at urban sites. Stream temperature surges were significantly correlated to urban land use, downstream distance, and discharge ($p = 0.02$).

Studies identified an urban micro-climate gradient and an urban heat island (UHI) effect in the city of Columbia, and noted that strategically located urban forest patches can be used to optimize localized cooling, carbon storage and cycling [95]. Similarly, floodplain work indicated that bottomland hardwood forest soils in Hinkson Creek Watershed store larger amounts of carbon relative to non-woody floodplain sites in the urban environment [96,97]. This information was not only useful in CAM discussions, and for local restoration policies, but also in current management discussions regarding the potential for bottomland hardwood forest restoration to meet carbon sequestration targets globally. Moreover, results repeatedly and conclusively supported the reestablishment of floodplain forests, where practicable, for the conservation of both groundwater and surface water quality. Studies showed that floodplain forests reduce subsurface shallow groundwater temperature fluctuations [98], can accept and thus process significantly ($\alpha = 0.05$, approximately 120 mm yr$^{-1}$) more water to storage than agricultural or grassland areas [99], significantly increase soil infiltration and soil volumetric water content holding capacity [100], increase consumptive use by vegetation [58], and improve freshwater routing, water quality, aquatic ecosystem conservation, and flood mitigation in mixed-land-use watersheds [98,101–103].

Program results also highlighted the effects of agricultural practices, specifically in the upper watershed, on the hydrologic regime of Hinkson Creek. For example, Zeiger and Hubbart [56] reported high concentrations of suspended sediment in the upper watershed, related to agricultural land uses. Similarly, Kellner and Hubbart [83] found indications of poor water quality in the agricultural upper watershed, as illustrated by levels of dissolved oxygen and pH values outside the recommended range for aquatic biota [91,92]. Such results suggest water quality and aquatic habitat degradation in Hinkson Creek is not limited to the activities and spatial extent of the city of Columbia, but rather is a complex watershed-scale issue involving integrated anthropogenic and natural processes. Similarly, a physical habitat assessment (PHA) showed that Hinkson Creek is altered by agricultural and urban land uses [104,105] that have also impacted macroinvertebrate assemblages in Hinkson Creek. This information was important in CAM discussions, considering macroinvertebrates are key species indicating general aquatic ecosystem status [55,106]. Results from the PHA clearly identified agricultural and urban land use alterations to channel geomorphology [105]. Results also showed increased substrate embeddedness (e.g., 80% vertical embeddedness of pool habitats) in the agricultural headwaters and in the lower urbanized reaches of Hinkson Creek [105]. The PHA assessment also revealed an increased frequency of fine streambed sediments coupled to increased substrate embeddedness in urbanized reaches. These results are in agreement with sediment studies in HCW that showed increased suspended sediment concentrations and increased fine suspended sediment particles in urban reaches [87–89].

Long-term multi-constituent datasets collected across a rural-urban land use gradient during the study included wet, average, and dry water years, and thus provided a distinct opportunity to assess the Soil Water Assessment Tool (SWAT). Results indicated "satisfactory" (Nash-Sutcliffe efficiency (NSE) values greater than 0.5) estimates of streamflow in Hinkson Creek during successive wet years [107]. The SWAT model also produced satisfactory estimates of monthly streamflow without model calibration [108]. However, uncalibrated SWAT model estimates of monthly sediment, total phosphorus, nitrate, nitrite, ammonium, and total inorganic nitrogen were unsatisfactory with NSE values less than 0.05. Model calibration at nested gauging sites increased NSE values above the aforementioned "satisfactory" threshold. The SWAT model was also used to simulate daily stream temperature with satisfactory results in Hinkson Creek [54]. Results identified useful model applications, including forecasting future hydrologic responses to urban growth and climate change [109–111], and pre-settlement land use model assessment [68,69]. Sunde et al. [109–111] simulated potential hydrologic consequences of increased impervious surfaces and climate change in HCW. For example, Sunde et al. [109] simulated three impervious growth scenarios using the

Imperviousness Change Analysis Tool (I-CAT) in HCW, and utilized climate change modeling results from the Coupled Model Intercomparison Project—Phase 5 (CMIP5) multimodel ensemble [110]. The simulated impervious growth and climate change data were used as model forcing's in SWAT to quantify the influence of projected impervious growth and climate change on water balance components. Collectively, results highlight the potential for combined and competing influences of climate change and development to result in decreased annual streamflow (−6.1%), and increased evapotranspiration (3.9%) in HCW [111]. The SWAT model was also used to simulate pre-settlement hydrologic conditions in HCW. Results confirmed the potential for agricultural and urban land-use influences on ecologically relevant daily streamflow regime metrics (streamflow magnitude, frequency, duration, timing, and rate of change) [68], and pollutant loading [69] in HCW. Critically, results indicated restoration of historic (i.e., pre-settlement) streamflow regimes are not fiscally obtainable targets in HCW and similar watersheds, where past and present land uses have extensively altered watershed hydrology and pollutant loading processes. This information is in agreement with the current understanding of environmental flows [41]. Ultimately, modeling results emphasize the great utility of the experimental design in advancing predictive potential and improving the accuracy of management practices.

### 2.4. Identified Unrecognized and "Unknown" Sources of Impairment

While anthropogenic pressures such as land use practices can exert driving influences on hydrologic and pollutant transport regimes, natural processes and landscape characteristics can compound impacts and confound the attribution of simple causal relationships to observed effects. For example, abnormal spatiotemporal streamflow relationships alerted the program director (Dr. Jason Hubbart) to possible (previously unidentified) hydrologic sink/source behavior in the upper-watershed [112]. Subsequent research uncovered archival evidence of historical subsurface coal mining, which may provide at least a partial explanation. Additional investigation identified hydrologic processes associated with natural landscape evolution, noted by early-20th-century researchers, which, when considered in the context of recent works, provide compelling alternative explanations for water quality and flow regime observations. Despite best-intentioned management, regulatory agencies, scientists, and local decision-makers did not account for such legacy practices and processes and instead relied on recent urban development as the proximate cause of designated impairment. Therefore, it is likely that historical land-use (coal mining) and landscape processes comprise cumulative, yet often unconsidered effects that contribute systemically to the observed hydrologic regimes of contemporary developing watersheds. In this regard, findings in HCW hold important implications for contemporary watershed management and suggest rethinking the case-by-case appropriateness of federal and state water impairment listings, and the achievability of restoration requirements therein.

Similarly, an investigation of the spatiotemporal variability of suspended sediment particle size class distribution (PSD) showed that the parameter best explaining the spatial pattern of PSD was not land use, but rather the surficial geology of the watershed [59]. The spatial pattern of surficial geology in the watershed (e.g., bedrock depth/constraints) also explained observations regarding suspended sediment concentrations [77,82], and stream geomorphology [81]. Finally, evidence was found to support the observation that the natural evolution of the Hinkson Creek hydrologic system is a contributing factor to observed water quality and stream geomorphology trends [81,112]. Specifically, historic Missouri River (confluence located approximately 8 km downstream) head-cutting and back-watering processes, at least in part, explain both channel incision and suspended sediment particle size characteristics in the lower watershed [81,112]. Notably, in conceptualizing the condition, management, and potential restoration of Hinkson Creek, the contribution of natural factors has often been overlooked in favor of a focus on anthropogenic disturbance [112]. However, these studies showed that a proper accounting of all contributing factors is required for accurate descriptions of system function and effective management [112].

## 3. Discussion

*Synthesis and Implications*

Synthesized salient, emergent results (i.e., "takeaways") of the work conducted during the HCW program include, (1) anthropogenic land use in HCW, including urban/suburban development and agriculture, is a primary driver of water quality degradation in Hinkson Creek; (2) land use practices impact suspended sediment characteristics and dynamics in HCW, including the flux of fine particles, which disproportionately contribute to water quality and aquatic habitat degradation; (3) streamflow alterations due to urban/suburban development result in increased streambed and bank erosion in Hinkson Creek, which increases sediment transport and disrupts aquatic habitat; (4) considering spatiotemporal water quality trends in Hinkson Creek, including dissolved oxygen levels, chloride concentrations, pH, water temperature, and suspended sediment concentrations, it is reasonable to expect stress conditions for aquatic biota throughout the stream, not only in urbanized/developed reaches; (5) several natural processes and watershed characteristics, such as surficial geology and stream system evolution, are likely compounding observed water quality and aquatic habitat degradation; and (6) given the varied and complicated set of factors contributing to water quality and aquatic habitat degradation in HCW, restoration of Hinkson Creek is likely subject to physical limitations and should be conceptualized in the context of achievable goals/objectives.

Results of the program highlight the compounding impacts of land use practices, hydroclimatic variability, and physical watershed characteristics on the suspended sediment, streamflow, and water quality regimes of Hinkson Creek. Land-use-driven alterations to the streamflow regime (e.g., increased runoff and flow magnitude, advanced peak hydrograph) of Hinkson Creek have resulted in increased pollutant transport and loading, and disturbance of aquatic habitat (bed incision, bank erosion, elevated stream temperature) that disrupts the biological integrity of the aquatic ecosystem. Restated, anthropogenic activities in the watershed exacerbate ecosystem vulnerabilities. Due to the many investigations concluded by the program, a more detailed and comprehensive description of the system is now available to stakeholders and decision-makers, which can be subsequently used to improve the management of the watershed [59,64,78,80,81,83,112]. The program also provides valuable insights regarding potential successes and challenges faced by collaborative adaptive management programs. Three issues emerged that should be emphasized to improve future CAM applications. First is the integrated approach with multiple objectives and multiple beneficiaries. For sustainable management, environmental factors need to be equally, if not more, greatly emphasized, relative to the economic aspects of project implementation. Second, local stakeholders must be involved as much and as early as possible. Local knowledge, gained by time and experience is critical for stakeholder buy-in, and project implementation and success. Ultimately, understanding values and opinions held by local communities is of critical importance [21,113], and stakeholders should be encouraged, and provided opportunities, to volunteer as team members to engage in the process [114]. Third, there must be regular updates and improvements to the plan. Given the inherent complexity of natural ecosystems, it is not surprising that effective resource management is dynamic, characterized by ongoing updates and refinements [21,115]. Collectively, more scientific and socioeconomic information and effective involvement of stakeholders are the primary components of collaborative adaptive management that lead to improved management decision-making [116–119].

## 4. Conclusions

Assuming human-induced land use change and in-tandem environmental change continue as expected, there is a need for streamlined (and normalized) collaborative adaptive management efforts to continuously monitor and respond to anthropogenic and natural pressures on water resources. Results from the experimental watershed approach and CAM processes highlighted here show the value of integrating knowledge and science with experience and the perspectives of scientists, stakeholders, and managers for more effective management decision-making. This is critical because, in the absence

of adequate observed data, sources of impairment are often unrecognized and/or listed as "unknown" in contemporary mixed-use watersheds. Additionally, sources of impairment can shift over time due to the combined influences of land use change, human population growth, and changing environmental conditions. Results from the case-study presented here clearly show that the experimental watershed study design can be used to provide science-based information critically needed to make informed management decisions in contemporary mixed-use watersheds. The design has the potential to be systematically applied in any watershed, thereby normalizing and standardizing study designs across watershed systems. In so doing, comparable inter- and intra-watershed information is collected, broader (multi-watershed) practices are implemented, and multi-scale costs are driven down over time.

In the Hinkson Creek Watershed in the Midwest USA, long-term monitoring of hydroclimate variables, streamflow, and multiple water quality constituents in nested sub-basins provided answers to specific questions generated during the CAM process. Additionally, hydrologic data collected and analyzed informed regional managers and advanced policy and science via generation of over 50 peer-review publications and 21 graduate student theses and dissertations. Key findings from the program showed (1) legacy effects, urban/suburban development and agriculture are primary drivers of alterations to watershed hydrology, streamflow regimes, multiple water quality constituents, and physical habitat degradation in Hinkson Creek; (2) several natural processes and watershed characteristics, such as surficial geology and stream system evolution, are likely compounding observed water quality and aquatic habitat degradation; and (3) given the varied and complicated set of factors contributing to water quality and aquatic habitat degradation, restoration of many USA CWA 303(d) listed streams and rivers like Hinkson Creek are likely subject to physical and fiscal limitations and should be conceptualized in the context of achievable goals/objectives. To this end, the nested-scale experimental watershed monitoring approach has served as a scalable model for studying natural and anthropogenic influences on water quantity, water quality, and stream physical habitat in contemporary mixed-use watersheds.

**Author Contributions:** conceptualization: J.A.H.; formal analysis: J.A.H., E.K., and S.J.Z.; investigation: J.A.H.; data curation: J.A.H.; writing-original draft preparation: J.A.H., E.K., and S.J.Z.; writing-review and editing: J.A.H., E.K., and S.J.Z.; supervision: J.A.H.; project administration: J.A.H.; funding acquisition: J.A.H.

**Funding:** The Hinkson Creek Experimental Watershed Program was founded and Directed by Jason Hubbart from 2007 to 2016. Funding was provided by the Missouri Department of Conservation and the U.S. Environmental Protection Agency Region 7 through the Missouri Department of Natural Resources (P.N: G08-NPS-17) under Section 319 of the Clean Water Act. Additional funding was provided by the partners of the Hinkson Creek Watershed Collaborative Adaptive Management program, the National Science Foundation under Award Number OIA-1458952, the USDA National Institute of Food and Agriculture, Hatch project 1011536, and the West Virginia Agricultural and Forestry Experiment Station. A glowing acknowledgment is due to John Schulz for support and guidance early in the program. Results presented may not reflect the views of the sponsors, and no official endorsement should be inferred. Finally, greatest thanks go to the many students, graduates, scientists, and contributors that devoted their work and time to the program.

**Conflicts of Interest:** The authors declare no conflict of interest.

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
