# Peer review of "A Case-Study Application of the Experimental Watershed Study Design to Advance Adaptive Management of Contemporary Watersheds"

_water, doi:10.3390/w11112355_

Round 1

Reviewer 1 Report

Interesting topic. My overall impression is that the article is too long (extremely wordy). I suggest to reduce the length of it. The introduction (section 1 is way too long). Also, the manuscript will benefit with a discussion section.

Author Response

Reviewer #1

Interesting topic. My overall impression is that the article is too long (extremely wordy). I suggest to reduce the length of it. The introduction (section 1 is way too long). Also, the manuscript will benefit with a discussion section.

Author Response:

We appreciate this constructive and instructive comment by the reviewer. We have thoroughly reviewed the manuscript, we carefully reduced length, and the manuscript is thus less wordy. We respectfully submit that while perhaps longer and more arduous in prose than the reviewer would prefer, we feel that given the highly varied topic(s) addressed in the article, there is justification for greater dedication to thorough articulation in places that require context. Regardless, we have indeed heeded the reviewers recommendation by reducing length where possible (reduced by approximately 1000 words), included a discussion section, and we hope the reviewer agrees the article is now acceptable for publication.

Reviewer 2 Report

The manuscript is well-written and in my opinion, deserves publication in Water journal. However, I suggest to enhance and quote experiences in other countries/catchments in the introduction paragraph in order to increase the appeal at an international level. I suggest to see, for example, the works "Evaluating performances of green roofs for stormwater runoff mitigation in a high flood risk urban catchment", "A procedure for designing natural water retention measures in new development areas under hydraulic-hydrologic invariance constraints", Exploring the performances of a new integrated approach of grey, green and blue infrastructures for combined sewer overflows remediation in high-density Urban areas". 

Author Response

Reviewer #2

The manuscript is well-written and in my opinion, deserves publication in Water journal. However, I suggest to enhance and quote experiences in other countries/catchments in the introduction paragraph in order to increase the appeal at an international level. I suggest to see, for example, the works "Evaluating performances of green roofs for stormwater runoff mitigation in a high flood risk urban catchment", "A procedure for designing natural water retention measures in new development areas under hydraulic-hydrologic invariance constraints", Exploring the performances of a new integrated approach of grey, green and blue infrastructures for combined sewer overflows remediation in high-density Urban areas".

Author Response:

We appreciate this reviewer’s positive comment, as well as constructive and instructive comments. We have followed the reviewer suggestion(s) and included additional articulation construct accordingly. We respectfully submit that examples such as green rooves is appropriate, however, there are a myriad of additional mitigation techniques that deserve equal billing in many (other) places globally. Accomplishing this endeavor is well beyond the scope of the current article, but would be very appropriate for a follow-on literature review. Regardless, we have heeded this reviewer recommendation, included recommended citations (plus) and we hope that the reviewer now agrees that the article is now acceptable for publication.

Reviewer 3 Report

First the all I would like to thanks Authors for hard work and a very interesting article. In my opinion, the conducted study is essential in the point of view in the urban catchments. Below is a few comments, which I believe, could a little bit improve the work.
1. Keywords: some keywords could be replaced, e.g. “Watershed Management” with “Urban Watershed Management”. Also the “Experimental Watershed” is not necessary.
2. The Introduction is definitely too long. This chapter should not be divided by the subchapters. Please provide in only the most important information about CAM approach (most important assumptions, examples of implementations).
3. In Introduction the Authors mentioned that CAM have been developed in the USA and Australia. What is the Author’s contribution, the novelty in reflect to approach? Is it the implementation in watershed management? If yes it should be clearly emphasized in the Introduction.
4. Also in the Introduction, the main aims of work must be clearly defined.
5. Case study: there is not necessary to repeat the references [40] at the end of the almost sentence. It could be placed at the end of the paragraph.
6. What is the source of the catchment land use information?
7. The second chapter should be divided into the two: study area characteristic and the results/
8. The Hinkson Creek Watershed is characterized in detail by physiographic factors. Also, this part should be completed by some information about the climatic conditions.
9. The Authors mentioned about some problems which could be met in the watershed. What is their main reason? The human activity or other?
10. The subchapter 2.1 – 2.4 should be separated as the results. Also, the information requires ordering. At first, the most important results, obtained from the CAM approach, should be presented, next the discussion.

Author Response

Reviewer #3

First the all I would like to thanks Authors for hard work and a very interesting article. In my opinion, the conducted study is essential in the point of view in the urban catchments. Below is a few comments, which I believe, could a little bit improve the work.

Authors Response: We appreciate this reviewer’s positive comment, as well as constructive and instructive comments. We have followed the reviewer suggestion(s) as per specific responses that follow. Thank you.

Keywords: some keywords could be replaced, e.g. “Watershed Management” with “Urban Watershed Management”. Also the “Experimental Watershed” is not necessary.

Author Response:

Thank you for these observations. We have made all recommended changes.

The Introduction is definitely too long. This chapter should not be divided by the subchapters. Please provide in only the most important information about CAM approach (most important assumptions, examples of implementations).

Author Response:

Thank you for this observation and recommendation(s). We agree that the Introduction was too long. However, we respectfully submit that while perhaps longer and more arduous in prose than the reviewer would prefer, we feel that given the highly varied topic(s) addressed in the article, there is justification for greater dedication to thorough articulation in places that require context in the introduction, including sub-headings that better frame the integrated approach. Despite this caveat, we have indeed heeded the reviewers recommendation by reducing length of the introduction where possible (reduced by approximately 1000 words), and we hope the reviewer agrees the article is now acceptable for publication.

In Introduction the Authors mentioned that CAM have been developed in the USA and Australia. What is the Author’s contribution, the novelty in reflect to approach? Is it the implementation in watershed management? If yes it should be clearly emphasized in the Introduction.

Author Response:

We appreciate this observation and submit that the author(s) contribution to the approach are multiply cited throughout the Introduction (and elsewhere). We further feel that to draw out explicit (by name or deed) examples of the authors contributions would be to draw too much attention to the author(s) which would detract (and create bias) from the equally important of other’s work cited throughout.

Also in the Introduction, the main aims of work must be clearly defined.

Author Response:

Thank you for this observation. We fully agree, and as noted above have thoroughly revised the introduction, including adding a clear objective statement to the end of section one of the introduction.

Case study: there is not necessary to repeat the references [40] at the end of the almost sentence. It could be placed at the end of the paragraph.

Author Response:

Thank you for this observation. We have made many of these changes in our revision. In other places we leave the end-of-sentence citations so that we are sure to be specific in our information for all readers. We ultimately agree that it is best to reduce redundancies where possible.

What is the source of the catchment land use information?

Author Response:

We appreciate this question/observation, and agree that a statement regarding the purpose of this information was necessary. We have added the following text to the beginning of the section. “To provide important context for the reader, in particular pertaining to transferability to other mixed (multi) use watersheds globally, we provide information about the case-study watershed used for the current work as follows.”

The second chapter should be divided into the two: study area characteristic and the results/

Author Response:

Thank you. Section 2 (Chapter 2) is divided as per recommendation.

The Hinkson Creek Watershed is characterized in detail by physiographic factors. Also, this part should be completed by some information about the climatic conditions.

Author Response:

Thank you for this observation. We agree and have added the following text, “Regional climate is dominated by continental polar air masses and maritime and continental tropical air masses during the winter and summer, respectively. Mean annual total precipitation is approximately 1096 mm, and mean annual air temperature is approximately 13.5 °C. A wet season occurs primarily from March through June.”

The Authors mentioned about some problems which could be met in the watershed. What is their main reason? The human activity or other?

Author Response:

We appreciate this comment. We have noted on pages 5 and 6 (and elsewhere) the reasons for CWA (polluted) listing, and the purpose of the study design to address those issues. Later we provide examples of findings that address the issues, and provide direction ranging from critical areas for mitigation to reasonableness of restoration. We hope this addresses the reviewer comment.

The subchapter 2.1 – 2.4 should be separated as the results. Also, the information requires ordering. At first, the most important results, obtained from the CAM approach, should be presented, next the discussion.

Author Response:

Thank you for these recommendations. We have reordered and relabeled the sections and we have reduced and synthesized material to make it more linear in presentation and hopefully palatable to the reader.

Round 2

Reviewer 3 Report

The reviewed work was significant improved. All my concenrs were solved. I will be happy to see article in Water MDPI.